# A Comprehensive Review Exploring the Role of Bone Morphogenetic Proteins [BMP]: Biological Mechanisms

**DOI:** 10.3390/cimb47030156

**Published:** 2025-02-27

**Authors:** Akanksha A. Kalal, Satyajit Mohapatra

**Affiliations:** SRM Centre for Clinical Trials and Research, SRM Medical College Hospital & Research Centre, SRM Institute of Science and Technology (SRMIST), Kattankulathur 603203, India; akankshk@srmist.edu.in

**Keywords:** bone morphogenetic proteins, cancer, TGF-β, SMAD

## Abstract

Bone morphogenetic proteins (BMPs) belong to the TGF-β family. They perform diverse roles in development, osteogenesis, and vasculogenesis. BMPs have crucial functions in embryonic development and regulate the specialization of various cell types. The dysregulation of BMP activity at various stages in signal transduction is associated with a diverse range of human diseases. It is not surprising that BMPs also have a role in tumor formation and control the progression of cancer through different phases. Nevertheless, their specific roles remain ambiguous and the findings regarding this have been inconsistent. The objective of this review is to highlight the important functions of BMP ligands, receptors, and signaling mediators and the subsequent effects on final cellular responses resulting from these signaling modalities. This review elucidates the dysregulation of BMPs identified in various cancer types, which serves as a predictive sign for favorable results in cancer therapy. Alterations in the BMP pathway can represent a crucial milestone in the genetic and molecular mechanisms that facilitate cancer formation. This review has shown that alterations in certain components of the BMP pathway are evident in various tumor forms, including breast, gastric, colorectal, and myeloma cancer. This review reinforces the conclusion that BMPs exert both beneficial and detrimental effects on cancer biology. Collectively, these findings indicate that BMPs serve multiple functions in cancer; therefore, directing therapeutic efforts to focus on BMP may be a highly effective method for treating several cancers.

## 1. Introduction

### 1.1. Bone Morphogenetic Proteins (BMP)

BMPs are cytokines belonging to the TGF superfamily and are produced and released as giant precursor proteins. These precursor proteins undergo proteolytic processing, which results in the release of their bioactive domain situated at the carboxy-terminus. The range of biological roles attributed to BMPs has significantly expanded since their initial discovery. Similarly, to other members of the TGF-β family, BMPs are versatile proteins that work in a way that depends on the specific circumstances. They serve as morphogens during early development, guide the construction of organs, and regulate tissue stability. The BMP family has significant functions in various processes during embryonic development and adult homeostasis. They regulate and commit to a cell lineage, as well as the morphogenesis, differentiation, proliferation, and apoptosis of different kinds of cells all through the body [1]. BMPs were first identified as substances that stimulate the abnormal development of cartilage and bone in rats. Subsequent analysis revealed that BMPs have several functions in skeletal growth, bone maintenance, and tissue regeneration through activating signal transduction through a complex consisting of specific trans membrane serine/threonine kinase receptors, namely BMPR1 and BMPR2. Furthermore, BMPs exhibit strong osteogenic properties, allowing for the production of new bone in living organisms [2].

BMP2, BMP4, BMP6, BMP7, and BMP9 are linked to elevated osteogenic activity [3]. BMP2 is a crucial element for bone formation and is currently being extensively researched for its potential use in context of medical treatments for humans, such as bone regeneration and therapeutic trials linked to bone pathogenesis [4]. BMP2 plays a significant role throughout the course of endochondral bone growth and stimulates the expression of markers associated with osteoblastic differentiation, such as alkaline phosphatase (ALP), osteocalcin, and RUNX2 [5]. Furthermore, BMP4 and BMP7 are heavily influenced by the vital contribution of endochondral bone growth and regeneration [6]. The commencement of this process necessitates the presence of BMP5. Normal skeletal development is facilitated by the combined action of BMP9 and vascular endothelial growth factor A (VEGFA), which efficiently promotes the production of bone in abnormal locations [7]. On the other hand, investigations conducted on mice with a genetic modification that eliminates BMP3 demonstrated that BMP3 functions as an inhibitory regulator of bone formation [8].

This review investigates the significance of BMP receptor signaling, BMPR2, and its dual involvement as a promoter and as an inhibitory function in various cancers, vascular development, and osteogenesis.

### 1.2. BMP Receptor Signaling

BMPs are either homo- or heterodimers that bind to specific heteromeric complexes consisting of two type I and two type II receptors. They exhibit little binding affinity towards either type I or type II receptors alone, but their affinity is enhanced when interacting with the type I–type II heteromeric complex [9]. The receptors consist of a compact extracellular domain, a transmembrane domain, an intracellular domain with serine/threonine action, and a cytoplasmic C-terminal tail part. Seven type I and three type II receptors are present. Type I receptors (activin receptor-like kinase-1 or ALK) have the ability to bind to both BMPs (ALK1, ALK2, ALK3, and ALK6) and TGF-β (ALK5). In contrast to TGF-proteins, BMPs have the ability to attach to type I receptors even when type II receptors are not present. Among the three type II receptors (the BMP type II receptor (BMPRII), the activin type II receptor (ActRII), and the activin type IIB receptor (ActRIIB)) in mammals. BMPR2 has a unique affinity for BMPs, while the other two receptors operate as binding sites for BMPs, activin, and myostatin [10].

BMPs are documented to stimulate the formation of bone and cartilage, demonstrating a diverse array of biological effects on multiple cell types. BMPs are crucial in embryonic and postnatal development, since they regulate cell differentiation, proliferation, motility, and survival, and hence preserve homeostasis in various organs and tissues [11]. In addition to its effects on carcinogenesis, BMP signaling plays a role in invasion and migratory activities, which are essential for metastatic dissemination. BMPs markedly enhance tumor migration by influencing the extracellular matrix (ECM) milieu, including integrins and matrix metalloproteinases (MMPs), which are critical determinants in tumor migration [12]. Research has indicated that BMPs exhibit noticeably elevated expression in tumors, and have been employed as novel biomarkers for cancer patients’ prognosis. BMPs may be considered an oncogene because of their correlation with the growth, differentiation, and apoptosis of cancer cells. We cannot simply categorize BMPs as oncogenes or tumor-suppressor genes because they are described as both stimulators and inhibitors in various cancers. All of the evidence mentioned above suggests that the tumor microenvironment and cell types affect how BMP signaling affects tumor growth. Therefore, we reviewed recent research on the bilateral effects of BMP in tumorigenesis and the underlying signaling pathways controlling the counterintuitive dilemmas in the current study.

Like more individuals belonging to the TGF-β family, BMPRI, serves as a substrate for BMPRII. BMPRII phosphorylates BMPRI at certain serine/threonine residues within GS-domain [13]. The negative regulator FKBP12 binds near the GS domain and, when it binds, it protects the serine/threonine residues from being phosphorylated by the type II kinase. Hence, the interaction between FKBP12 and BMPRI establishes a specific level of initiation, so that when type I and II receptors come together on the cell surface without a ligand present, an intracellular signal is not instantly triggered [14]. Figure 1. When BMPRI is activated, it triggers intracellular signaling by activating receptor-regulated R– SMADs, specifically SMAD1, SMAD5, and SMAD8. TGF-β cytokines and activins often trigger the phosphorylation of R-SMAD2 and SMAD3 in the majority of cells, distinguishing them from the current subject. Antibodies that specifically target phosphorylated forms of Smad proteins, such as p-SMAD1/5/8 and p-SMAD2, are developed and commonly employed to assess the level of cell and tissue receptor activation [15].

R-SMADs become active by forming heteromeric complexes with Co-SMAD4. These complexes work together with transcription factors, transcriptional co-activators, and repressors to manage certain gene transcriptional responses. R-SMADs and SMAD4 have an affinity for sequences that are rich in guanine and cytosine and for DNA sequences that contain the CGTA sequence, respectively. ID1 is a well-known target gene for BMPs. The gene promoter was found to contain BMP-SMAD-sensitive regions, which were then cloned together to create a very effective transcription reporter called the BRE-luc reporter [16]. In addition to the classic SMAD pathway, BMPR activation can also trigger non-SMAD signaling, which involves the activation of p38, JNK MAP kinases, and small GTPases like Rho and Rac within the cell [17].

A homeobox gene called Tlx-2 is expressed in mouse embryos’ primitive streak. This is a crucial downstream target of BMP signaling that is necessary for mouse gastrulation. The activation of the Tlx-2 promoter by BMP signaling is facilitated by the BMP type I receptors, specifically ALK3 or ALK6, in conjunction with Smad1. Mammalian mesoderm production depends on the BMP/Tlx-2 signaling pathway, which also provides a mechanistic explanation for how TGFβ-related proteins work during this developmental stage [18]. Muscle segment homeobox (Msx2), a transcription factor with a homeobox, is a member of the highly conserved and extensively expressed msh superfamily. BMP signaling actively modulates the expression of Msx2, which is integral to numerous developmental processes, including osteogenesis, odontogenesis, and limb patterning. This regulation occurs through the activation of Msx2, which governs cellular differentiation and proliferation. Consequently, Msx2 serves as a pivotal mediator of BMP signaling across various tissues [19]. Dlx5 is a transcription factor that promotes bone formation and is expressed during the later stages of osteoblast differentiation. The expression of Dlx5 is generally detected in particular neuronal tissues, as well as in the developing skeletal structures, which encompass cartilage, bone, and teeth. Dlx5 functions as the principal target of BMP signaling. During the differentiation of osteoblasts induced by BMP-2, Dlx5 plays a pivotal role in the activation of the downstream osteogenic master transcription factor Runx2. These factors function either sequentially or in conjunction to enhance the expression of bone marker genes that are indicative of trans differentiation [20]. Consequently, in this investigation, we examined recent research concerning the bilateral effects of BMP in carcinogenesis and the signaling pathways that govern the associated contradictions.

### 1.3. BMPR2

*BMPR2* consists of 13 exons, coding for four domains, the extracellular, Trans membrane, kinase, and Cytoplasmic domains (Figure 2). Type II receptors differ from type I receptors as they contain extended 509 amino acid Carboxyl terminal domain encoded exons 12 and 13, followed by the kinase domain. The BMPR2 is a receptor belonging to the TGF-β superfamily. It is found in many different tissues and organs, such as the pulmonary vascular endothelium, pulmonary vascular smooth muscle, cerebellum, hippocampus, heart, liver, pancreas, and kidney. The BMPR2 receptor has a short and long form, either with or without the C-terminal tail. The interaction of the C-terminal tail with the proteins in cytosols alters pathways such as the mitogen-activated protein kinase (MAPK) and SMAD pathways [21].

### 1.4. Role of BMPR2

#### 1.4.1. Vascular Development

BMP signaling promotes endothelial specification, subsequent venous differentiation, and angiogenesis during embryonic development, hence sustaining the vascular system [22]. BMPR2 is not highly expressed in the smooth muscle of the airways and arteries. However, its expression is very visible in the endothelial cells and smooth muscle layer of blood vessels in the lungs of healthy individuals. A majority of familial Pulmonary arterial hypertension (PAH) patients, over 70%, and a significant proportion of idiopathic PAH patients, over 20%, possess heterozygous mutations that impair BMPR2 activity. Individuals diagnosed with PAH with *BMPR2* mutations demonstrate an intensified form of the disease with a higher likelihood of mortality compared to those who do not have *BMPR2* mutations [23]. Mutations in *BMPR2* in PAH patients are linked to high pulmonary arterial pressure and poor prognosis. In addition to mutations in *BMPR2*, other genes that encode for components involved in BMP signaling, such as GDF-2, ACVRL1, ENDOGLIN, and SMAD8, have been linked to the development of PAH [24,25,26,27,28]

#### 1.4.2. Osteogenesis

BMPR2 contributes to osteoblast differentiation and serves as significant mediator in the production of bone and skeletal development in various situations, such as fracture healing, otospongiosis, and osteosclerosis [29,30]. The differentiation and maturation of osteoblasts involves the development of mesenchymal stem cells (MSCs) into osteoblast progenitors. Further, osteoblast progenitors undergo maturation into osteoblasts, which exhibit different phenotypes involved in the formation of bone [31]. Katagiri et al. discovered C3H10T1/2 cells have the ability to transform into osteoblast-like cells when exposed to recombinant human BMP2 protein. In addition, the researchers confirmed the osteogenic activity of BMP2 by conducting experiments on myoblastic C2C12 cells. The administration of BMP2 suppressed the formation of myotubes. Alternatively, the cells initiated ID1 expression to facilitate the differentiation of myoblasts into osteoblasts in response to BMP2 [32]. Wu et al. demonstrated that both BMPR2 and ACVR2 play a role in the osteogenic differentiation of C3H10T1/2 when treated with BMP9. All these studies indicate their elucidative role in skeletal development [33].

Since the identification of BMPs by Urist in 1965, followed by their molecular cloning and characterization, these proteins have been successfully integrated into clinical practice for particular indications [34]. The overarching function of BMPs in the processes of bone formation during both the developmental stages and the repair of fractures has been extensively researched. Their osteoinductive capabilities have been substantiated through their application in ectopic sites as well as in critical-sized bone defects. Currently, recombinant technology is employed to mass-produce human BMPs for clinical applications. These recombinant human BMPs (rhBMPs) of BMP-2 and BMP-7 are currently employed in the fields of dental tissue engineering, fracture healing, and spinal fusion. rhBMP-7 is a protein utilized in the treatment of bone fractures. In the study of Bonato et al. the initial affirmative hypothesis was substantiated, revealing that a greater volume of bone density was observed in thin-coated implants associated with rhBMP-7, particularly in the regions surrounding implants positioned in over-instrumented sites [35]. Comparable findings were reported by Nemcakova et al. who determined that BMP-7 facilitated accelerated osseointegration and enhanced bone healing, thereby indicating potential for earlier loading. Moreover, Schierano et al. determined that rhBMP-7 enhanced osteogenic and anti-inflammatory properties when applied to surface-coated implants [36].

#### 1.4.3. In Cancer

BMPR2 has a multifaceted role in cancer. It functions as a tumor-suppressor gene in multiple forms of cancer, such as colorectal, breast, ovarian, and prostate cancer [Figure 3]. Loss-of-function mutations or the decreased expression of BMPR2 are linked to heightened tumor proliferation, infiltration, and spread to other parts of the body [37]. The BMPR2 signaling pathway hinders cell proliferation, triggers apoptosis, and controls epithelial–mesenchymal transition (EMT) in order to restrict the formation and spread of tumors. BMPR2 engages with multiple signaling pathways, including Wnt/β–catenin and TGF-β, and the disruption of these interactions can lead to the formation of tumors. The decreased presence or absence of BMPR2 is linked to a negative outlook in certain cancer types [38]. The precise function of BMPR2 can differ based on the particular type of cancer and the cellular environment. Further research is necessary to gain a thorough understanding of the processes and therapeutic implications of BMPR2 in cancer (Table 1).

Langenfeld et al.’s study revealed that A549 cells overexpressing BMP2 were administered into nude mice and discovered that BMP2-mediated signaling played a role in tumor angiogenesis [39]. Wiley et al. demonstrated that the signaling pathway involving BMP2 and BMPR2 controls the growth of new blood vessels from the axial vein during the development of zebrafish. Their study revealed that BMPR2-dependent signaling stimulates the multiplication of endothelial cells and angiogenesis [40].

BMPs have the ability to stimulate the transformation of normal brain stem cells into astrocytes [41]. Research has shown that BMP ligands inhibit the formation of glioblastoma Cancer Stem Cells (CSCs) [42,43,44]. Developmental studies have shown that BMPs can either stimulate the formation of new neurons or promote the differentiation of glial cells. The specific outcome depends on factors such as the stage of embryonic development, the origin of the cells, and the age of the target cells. In glioblastomas, the activation of BMP signaling results in the passiveness of glial CSCs, which are resistive to temozolomide chemotherapy and radiation. ID1 governs the processes of cell invasion, proliferation, and self-renewal in various types of cancer, including glioblastoma CSCs. ID1 enhances the tumorigenicity of GBMs and promotes the self-renewal of glial CSCs by suppressing the expression of BMPR2. The inhibition of BMP signaling with inhibitors markedly reduces the expression of ID1 without causing an increase in the expression of BMPR2. Joel Kaye research has indicated that the act of suppressing BMPR2 leads to a higher rate of cell death in cancer cells compared to inhibiting BMP type 1 receptors. The data described by Kaye et al. indicate that suppressing BMP signaling could be a promising treatment strategy for GBMs [45].

The BMP signaling pathway has a tumor-inhibitory effect, whereas the conventional TGF-β pathway is involved in the development of cancer. A study revealed that the *BMPR1A* gene is responsible for the onset of Juvenile Polyposis [46]. For example, there is a connection between the development of breast cancer (BC) and BMP signaling, although there are variations in the outcomes observed in human tissue compared to animal models. Owens and colleagues inhibited BMPR2 signaling in transgenic mice by increasing the expression of a mutant BMPR2-DelEx4-DN allele. The mice were genetically modified to produce a mouse mammary tumor virus polyoma middle tumor antigen, which led to the formation of mammary tumors. The inhibition of BMPR2 resulted in enhanced tumor cell proliferation, motility, and invasion, as well as the development of a more inflammatory tumor microenvironment [47].

As of 2020, BC has overtaken lung cancer to become the leading form of cancer. However, lung cancer still has the greatest fatality rate among all cancers [48]. The overexpression of BMP2 in BC patients has been linked to a lower likelihood of disease-free survival. This is due to the increased activation of the AKT/mTOR pathway. This association was shown in a study involving 272 patients by Wang et al. [49].

The role of BMP signaling in BC is varied. BMP receptors, as crucial downstream effectors of BMP signaling, have dual functions in cancer progression, serving as both promoters and inhibitors of cancer. A study demonstrated a correlation between decreased expression of BMPR1b and unfavorable prognosis, as well as the heightened proliferation of BC cells [50]. Various studies have presented conflicting findings about the roles of BMPR2 in BC. A study demonstrated that the interference of BMPR2 enhances the spread of BC [47], but another investigation revealed that the interruption of BMPR2 hinders the proliferation of BC cells. Pouliot et al. speculated that the most effective method for elucidating the role of BMPs in BC cells would involve the constitutive repression of the BMP signaling pathway. In pursuit of this objective, the engineered Dominant Negative Type II Bone Morphogenetic Protein Receptor II (DN-BMPRII) and developed T-47D cell lines consistently express the recombinant receptors. The findings indicate that both the transient and stable overexpression of DN-BMPRII effectively obstruct the activation of Smad1 by BMP-2 and subsequently inhibit the proliferation of BC cells. These findings indicate that BMPs interacting with BMPR-IIs play a significant role in the proliferation of human BC cells [51]. Another study conducted by Pouliot et al. demonstrated that BMP2 significantly suppressed the proliferation of BC cell lines that exhibit the expression of both Smad1 and Smad4. This inhibition of growth was found to be associated with the upregulation of both p21 mRNA and protein levels. The activity of the p21 promoter necessitated the presence of both Smad1 and Smad4 and was stimulated by either BMP2 or a constitutively active type I BMP receptor. Smad1, Smad4, and BMP2, as well as constitutively active BMP-2 type I receptors (ALK3QD and ALK6QD), function synergistically to enhance the activity of the p21 promoter in human BC cells [52].

The study conducted by Clement et al. demonstrated that increased levels of BMP-2 augment the tumorigenic characteristics of breast carcinoma cells, propelling these cells towards a more aggressive phenotype characterized by estrogen-independent growth. [53]. Recent evidence in the context of human BC has indicated that the overexpression of BMPs, particularly BMP4 and BMP7, is associated with the progression of the disease to more advanced stages [54]. Gosh et al. presented the initial evidence indicating that BMP-2 inhibits the estradiol-induced proliferation of human BC cells. The observed effect of BMP-2 seems to be facilitated through the suppression of positive regulatory proteins associated with the cell cycle. The hyperproliferation of estrogen-responsive BC cells constitutes a significant factor in the etiology of tumor formation during the initial phases of BC [55].

Prior studies have shown that BMPs, along with TGF-β, can enhance the invasion and bone metastases of BC in live organisms. Both BMP2 and TGF-β3 increased the movement of MDAMB-231 cells in both in vitro and xenograft model settings [56]. According to a study conducted by Buijs and colleagues, a reduced level of BMP7 expression in primary tumors is strongly associated with the occurrence of bone metastases in BC. BMP7 has the potential to impede the growth and specialization of BC cells both at the original tumor site and in the bone [57]. Further evidence supporting the role of BMPs in bone metastasis was obtained from another finding indicating that the BMP2/BMP7 heterodimer hindered the colonization of BC cells in bone [58]. All of these studies unequivocally demonstrate that BMPs are differentially expressed in BC, and that BC possesses the capacity for BMP signaling.

Chondrosarcoma is a malignant tumor that originates from cartilage and is the second most frequent form of primary bone cancer. Jiao et al. showed that Chondrosarcoma undergoes apoptosis and autophagy in response to BMPR2 suppression. Their results suggest that BMPR2 plays a vital role in the development of chondrosarcoma and can function as a valuable prognostic indicator for this type of cancer. Inhibiting BMPR2 could potentially provide a beneficial therapeutic approach for the treatment of chondrosarcoma [59].

The increased expression of BMPR2 was associated with a lower overall survival rate in individuals with ovarian cancer (OC). Fukuda et al. showed that both BMP2 and BMPR2 promoted the proliferation of OC cells, while inhibitors of the BMP receptor kinase hindered the growth of OC cells in both cultured cells and a mouse model. The study indicates that BMP signaling has the potential to promote tumor growth in OC. Consequently, the use of BMP inhibitors could prove beneficial as therapeutic treatments for patients with OC [60].

Neuroblastoma (NB) is the predominant extracranial solid tumor, found outside the brain in children, and it is the most often occurring form of cancer detected before the age of one [61]. The management of NB varies for each patient and is determined by the location of the main tumor, the histology of the tumor, and the presence of metastasis, which is observed in around 70% of patients at the time of diagnosis. The outlook for children with NB has significantly improved, primarily due to advancements in therapy for patients with low-grade tumors. Nevertheless, the survival percentage for people diagnosed with high-risk illness, characterized by high grades, is merely 20 to 40% [62].

Cui et al. revealed a considerable decrease in the expression of BMPR2 in primary neuroblastoma tissue, particularly in cases of high-grade illness. When BMPR2 is artificially increased in NB cells grown in a controlled environment, it prevents their growth and ability to form colonies. Conversely, reducing the levels of BMPR2 increases the growth of NB cells both in controlled environments and in living organisms. An examination of past data found that there is a correlation between elevated BMPR2 expression and improved prognosis in individuals with NB. Therefore, it seems that BMPR2 has a significant function in the development of NB, and activating the BMPR2 system could be a new approach for treating it [63].

Hepatocellular carcinoma (HCC) accounts for 80% of primary malignant liver tumors and has emerged as a major public health issue. Liver cancer is projected to increase by 55% during the next two decades, presenting a significant global concern [64]. The involvement of BMP in the occurrence of HCC has been extensively studied. Specifically, the increased expression of BMP4 is strongly associated with the presence of multiple tumor nodules, the TNM stage, and the invasion of blood vessels [65]. The overexpression of BMP9 in HCC cell lines can augment the process of EMT, which is a critical mechanism for initiating the invasion and spread of cancer [66]. While not solely caused by a lack of BMPR2, these difficulties may be important in specific patients who undergo BMPR2 restoration treatments. Correcting the imbalance of the TGF-β1/BMP7 requires a significant decrease in the growth and dissemination of HCC [67]. Another study indicates that BMP2 can stimulate the activation of myeloid-derived suppressor cells and the release of IL-6, thereby promoting the rapid increase in the number of cancer cells and facilitating the progression of liver cancer [68]. Li et al. study highlights the scientific importance of BMP2 as a regulator of tumor angiogenesis in HCC, which subsequently stimulates tumor development and metastasis. The findings were supported by in vivo experiments, which showed that increasing the expression of BMP2 within liver tumors in mice promotes tumor growth and the spread of cancer cells to the lungs. Conversely, suppressing BMP2 significantly reduces the size of the tumors and inhibits the growth of blood vessels, as indicated by the decrease in CD34+ areas [69].

Pancreatic ductal adenocarcinoma (PDAC) is a prevalent and aggressive form of cancer, characterized by a low 5-year survival rate that has shown little improvement despite advancements in surgical techniques and additional treatments over the years [70]. Wang et al. found that BMPR2 is overexpressed in PDAC tumors relative to normal pancreas tissues [71].

Gastric cancer (GC) is a prevalent kind of cancer that affects the digestive system and is responsible for a significant number of deaths caused by cancer globally; it is reported to be the second most prevalent reason for these fatalities [72]. The presence of BMP2 was more commonly detected in gastric tumors of the intestinal type compared to those of the diffuse type. This association is linked to the level of differentiation and the occurrence of lymph node metastases. The study demonstrates that BMP2 enhances the spread of GC via controlling the activities of NF-ĸB and MMP9 through the PI3K/Akt and MAPK pathways [73]. Sun et al. discovered that the expression of BMP5 in GC tissues was remarkably reduced in comparison to healthy tissues. Conversely, the levels of ACVRL1, ACVR1, TGFBR1, and BMPR2 expression were markedly increased. Furthermore, a survival study conducted on the TCGA and GEO databases revealed that individuals exhibiting elevated levels of BMPR2 experienced significantly reduced survival periods in comparison to individuals with low-expression tumors [74].

Colorectal cancer (CRC) ranks as the fourth most prevalent form of cancer, causing approximately 1 million fatalities annually at a global scale [75]. Bonjoch et al. showed that alterations in BMPR2 function have the ability to modulate the activation of the BMP pathway, perhaps resulting in complete loss of BMPR2 function and the subsequent elimination of cell division inhibition. Due to the significant importance of this pathway in the development of CRC, their study suggests that BMPR2 could be a promising candidate gene for assessing the risk of hereditary CRC [76].

Studies have shown that, in colon cancers (CC), BMPR1a and SMAD4 are mutated, suggesting that BMP-mediated signaling can act as a tumor-suppressor. Kodach et al. observed a deficiency in BMPR2 expression in microsatellite instable (MSI) CC cell lines, specifically HCT116, DLD1, SW48, and LOVO. However, BMPR2 expression was shown to be normal in microsatellite stable (MSS) CC cell lines [77].

Lung adenocarcinoma is the predominant form of cancer and the primary contributor to cancer-related fatalities among males and females globally. Between 2010 and 2016, the five-year relative rate of survival for lung cancer stood at a mere 21%, with the bulk of patient fatalities resulting from complications arising from metastases [78]. Kuei Wu et al. found that BMP2 is enhanced in lung adenocarcinoma patients who have lymph node metastasis in comparison to those who do not have it. The study unequivocally showed, through the use of an in vivo orthotopic mouse model, that BMP2 actively enhances the spread of lung adenocarcinoma to other parts of the body. Cell migration and invasiveness were drastically decreased when BMP2 or its receptor BMPR2 were depleted [79].

According to the findings, the expression of BMPR2 is considerably reduced in prostate cancer (PC) tissues. The decreased expression of BMPR2 demonstrated a statistically significant association with unfavorable prognosis, including cancer recurrence and lower 5-year survival rates [80]. Horvath et al. proposed that BMP2 could serve as an indicator of unfavorable prognosis, as a considerable decrease in PC is observed in comparison to in benign prostate tissue [81]. Lu et al. discovered that suppressing BMP signaling through the extinction of BMPR2 in Pten-deficient PC cells substantially increases the lifespan of mice, which could have significant practical application, as it indicates that targeting the BMP pathway may be a viable treatment strategy to enhance overall survival in a significant portion of PC patients [82].

The key molecules that contribute to the survival and expansion of multiple myeloma (MM) cells include anti-apoptotic members of the BCL2 family, together with several growth factors, adhesion molecules, and cytokines, including IL6 [83,84,85]. Various cytokines are documented to impede the proliferation of myeloma cells under various culture conditions, including BMPs. The BMPs intracellular pathway includes type I and II kinase receptors and SMAD (small mothers against decapentaplegic), which are intracellular signal transducers. The phosphorylation of the SMAD1/5/8 complex regulates target gene transcription, along with inhibiting the DNA binding (ID) gene family [86,87,88]. The predominant receptor found in myeloma cells is ALK2, which is produced by ACVR1. The study revealed a considerable overrepresentation of genes from the PI3K/AKT/mTOR cascade, suggesting that these genes play a favorable role in regulating ALK2-mediated apoptosis. The PI3K/AKT/mTOR pathway was documented to be overexpressed in various types of malignancies, including MM. The study revealed a notable deficiency of transcriptional regulators of TP53, suggesting that there is a negative control mechanism in place for ALK2-mediated apoptosis. Ultimately, the induction of cell death in MM cells by BMP is dependent on the conventional BMP signaling pathway [89].

The findings of Grcevic et al. suggest that BMPs enhance the survival of myeloma cells by activating the ID family of oncogenes and hence increasing the balance between pro-survival and pro-apoptotic molecules. The potential biological impact of BMPs in vivo may involve the stimulation of cell survival and the build-up of myeloma cells within bone marrow through an autocrine mechanism [90]. BMPR2 hinders the formation of active receptor signaling complexes, including ALK2, ACVR2A, and ACVR2B. As a result, it inhibits the activation of SMAD1/5/8 and the subsequent death of myeloma cells mediated by ALK2 [91].

**Table 1 cimb-47-00156-t001:** Displaying the dysregulation of BMPR2 and its effect on various cancer types.

Reference	Cancers	Mechanism of Action	Effect
Cui et al., 2017 [63]	Neuroblastoma	Downregulation of BMPR2 [ShRNA knockdown] with loss of BMPR2-mediated signaling	Increased cell proliferation and clonogenicity
Kaye et al., 2022 [45]	Glioblastoma	Inhibiting BMP pathway by suppressing BMPR2	Induced cell death
Wang et al., 2020 [49]	Breast cancer	Higher expression of BMP2 and activation of AKT/mTOR pathway	Micro-calcification and poor prognosis
Jiao et al., 2014 [59]	Chondrosarcoma	Silencing of BMPR2 promoted G2/M cell cycle arrest through caspase 3-dependent pathway	Induced chondrosarcoma cell cycle arrest
Lu et al., 2017 [82]	Prostate	BMPR2 and TGF-β through SMAD-independent and -dependent pathways	Suppressed prostate cancer development
Wang e al., 2020 [49]	Pancreatic	BMPR2 expression promoted G2/M phase through GRB2/PI3K/AKT signaling pathway	Promoted PDAC cell proliferation with worse overall survival.
Sun et al., 2020 [74]	Gastric cancer	Higher expression of BMPR2 through the regulation of tumor-associated angiogenesis and lymphangiogenesis	Facilitated spread of a tumor with poor overall survival.
Bonjoch et al., 2023 [76]	Colorectal cancer	Mutation in BMPR2 with reduced R-smad phosphorylation with modifications in the BMP pathway and activation of other non-canonical pathways, such as JNK, ERK, and p28	Increased proliferation.
Li et al., 2024 [69]	Liver [HCC]	BMP2 promoted metastasis and growth by increasing angiogenesis in HCC	Promoted growth and EMT with indication of worst prognosis
Wu CK et al., 2022 [79]	Lung adenocarcinoma	BMPR2 was overexpressed in lung adenocarcinoma metastasis through activation of the SMAD1/5/8 signaling pathway	Promoted cell migration and invasiveness
Fukuda et al., 2020 [60]	Ovarian cancer	BMP2 significantly enhanced OCs by inducing EMT	Enhanced proliferation and migration of OCs and poor prognosis
Olsen OE et al., 2018 [91]	Multiple myeloma	BMPs potently induce growth arrest through the BMP signaling pathway	Promoted apoptosis in myeloma cells.

### 1.5. The Dual Activity of BMPs

Currently, there is a heightened comprehension of the crucial roles that BMPs play in cancer. Collectively, the abundance of contradictory findings suggests that the same ligand can have varying effects depending on the type of cancer. Therefore, it is apparent that several members of the BMP family should not be treated as equivalent and should not be examined in the same manner. Moreover, identical BMP ligands may have varying effects on the same type of cancer, contingent upon the specific study. The conclusions drawn from a single cell line may be overly simplistic; hence, it is advisable to employ a variety of cancer cell lines or various types of tumors. The prevailing agreement is that BMPs may function as tumor-promoters and oncogenes in the onset of cancer.

### 1.6. Targeting BMP Signaling in Cancer Treatment: Challenges and Limitations

In cancer therapy, targeting BMP signaling poses considerable challenges due to its intricate and context-dependent characteristics, including its dual role as a tumor-suppressor and -promoter based on cancer type and cellular environment, the existence of various BMP ligands and receptors, and the likelihood of resistance mechanisms arising from pathway modifications or compensatory signaling pathways.

BMP signaling often overlaps with other signaling pathways, occasionally serving as a promoter of tumor metastasis. A study demonstrated its tumor-promoting effects in ovarian and endometrial cancer. In the uterine corpus, BMP2 is retained by both endometrial stromal cells and vascular endothelial cells. Consequently, BMP2 activates the endothelial cells [92]. In the micro-vascular network of human blood endothelial progenitor cells and lung tumors, BMP-2 and BMP-4 have generally been proposed to be angiogenic factors that promote tube formation and the neovascularization of melanoma cells by associating with VEFG to stimulate angiogenesis [93].

The activity of BMP signaling is meticulously regulated by numerous variables, and breaking this balance might modify the properties of normal cells, resulting in their transformation into tumor cells. In this context, the secreted antagonists are crucial in the regulatory framework of BMP signaling. The tumor microenvironment contains an abundance of different secreted substances, including antagonists of BMP signaling. The levels and function of BMP ligands and antagonists may rely on complex intercellular communication, and it has been proposed that cancer stem cells may release BMP signaling antagonists to suppress the BMP pathway in the tumor microenvironment. Therefore, examining the functions of BMP ligands and antagonists in the tumor microenvironment may yield significant insights into the regulatory networks that affect cancer genesis and progression. Despite the fact that the intricacy of BMP signaling in metabolic malfunctions, vascular diseases, and cancer is still not fully comprehended, there is increasing evidence from a variety of systems that BMPs have significant therapeutic potential.

## 2. Conclusions

Previous studies have demonstrated that BMP signaling in different types of malignancies activates pathways that promote cell survival. It is suggested that the development of BMP inhibitors as possible therapies for glioblastoma treatment is necessary, as BMP signaling is promotes the growth of glioblastomas. Dysregulated BMPs have been identified in numerous cancer types and serve as prognostic indicator for favorable outcomes in cancer therapy. Changes in the components of the BMP pathway might result in the buildup of cells that do not fully develop their characteristics because the signal that promotes differentiation is lacking. Therefore, it is evident that modifications in the BMP pathway can serve as a significant milestone in the genetic and molecular processes that contribute to the development of cancer. Studies have demonstrated that changes in certain elements of the BMP pathway are observed in several types of tumors, including prostate, colon, and BC. This review further affirms the overall finding that BMPs have both positive and negative effects in cancer biology. Their impact as either tumor-suppressors or tumor-promoters is contingent upon the specific tissue or cell type in the micro-setting, the patient’s epigenetic background, and the stage of tumor growth.

## Figures and Tables

**Figure 1 cimb-47-00156-f001:**
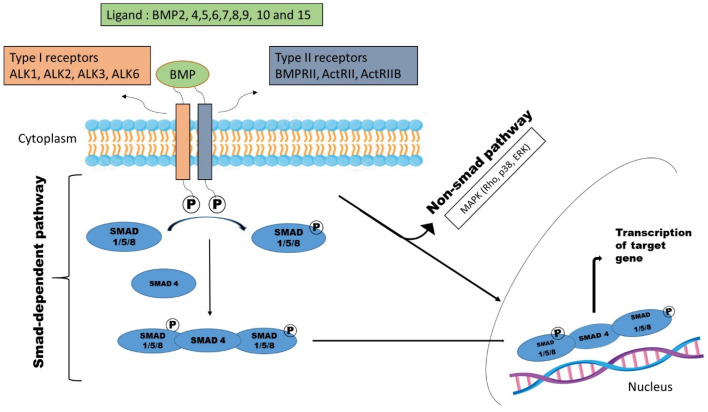
Schematic representation of BMP signaling pathway. Abbreviations: ALK—activin receptor-like kinase; BMP—morphogenetic proteins; BMPRII—BMP type II receptor; ActRII—activin type II receptor; MAPK—mitogen-activated protein kinase; ERK—extracellular signal-regulated kinase.

**Figure 2 cimb-47-00156-f002:**
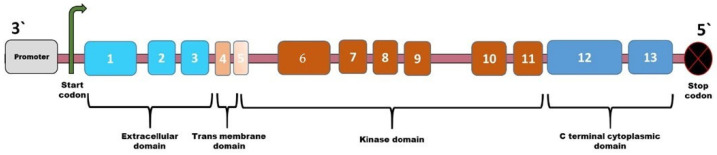
Schematic illustration of BMPR2 gene.

**Figure 3 cimb-47-00156-f003:**
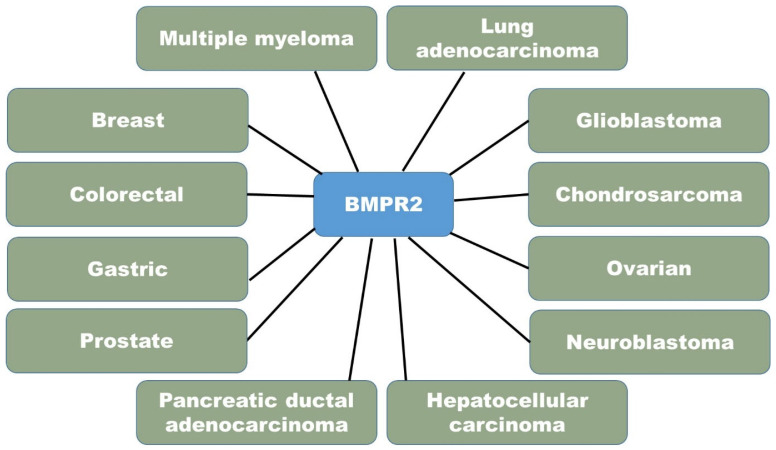
Association of BMPR2 in different cancer types.

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
