# Peer review of "A Comprehensive Review Exploring the Role of Bone Morphogenetic Proteins [BMP]: Biological Mechanisms"

_cimb, 2025, doi:10.3390/cimb47030156_

Round 1
Reviewer 1 Report
Comments and Suggestions for Authors
The manuscript provides a comprehensive review of the role of Bone Morphogenetic Proteins (BMPs) and BMP receptor signaling, with a particular focus on BMPR2. The review is well-structured and covers a broad range of topics, making this manuscript a valuable resource for biology researchers.
Strengths:
The manuscript thoroughly reviews the role of BMPs and BMPR2 in diverse biological processes, including embryonic development, bone formation, and cancer. The manuscript highlights the potential therapeutic implications of BMP signaling in cancer treatment, particularly in glioblastoma, breast cancer, and other malignancies. The discussion on BMP inhibitors and their possible use in therapy is timely and relevant.
The manuscript cites recent studies (up to 2024), ensuring that the review is current and reflects the latest advancements in the field.
Weaknesses:
1. the manuscript primarily summarizes existing studies without providing sufficient critical analysis or synthesis of the findings. For example, the contradictory roles of BMPR2 in breast cancer (lines 198-201) could be discussed in greater depth, with potential explanations for these discrepancies (e.g., differences in experimental models, cell types, or signaling contexts).
2. The manuscript could benefit from discussing the challenges and limitations of targeting BMP signaling in cancer therapy. For example, the potential for off-target effects or resistance mechanisms could be explored.
Specific Comments:
1. Introduction The introduction could be strengthened by briefly mentioning the clinical significance of BMP signaling, setting the stage for the later discussion on cancer.
2. Line 110 to line 118, BMPR2 and BMP2R are mixed used.
3. BMP Receptor Signaling (Section 1.2): The discussion of non-SMAD signaling (lines 90-93) could be expanded to include more details on the downstream effects of these pathways. ID1 is downstream of BMP signaling; however, other than ID1, other targets could be discussed in some tissue, for example, bone tissue: Msx2
4. BMPR2 in Cancer (Section 1.4.3): The contradictory findings regarding BMPR2 in breast cancer (lines 198-201) should be analyzed in greater depth, with potential explanations for these discrepancies.
Author Response
Comments 1: The manuscript provides a comprehensive review of the role of Bone Morphogenetic Proteins (BMPs) and BMP receptor signaling, with a particular focus on BMPR2. The review is well-structured and covers a broad range of topics, making this manuscript a valuable resource for biology researchers.
Response 1: Thank you for your positive feedback. We appreciate your recognition of the comprehensive nature and structure of our review.
Comments 2: Strengths: The manuscript thoroughly reviews the role of BMPs and BMPR2 in diverse biological processes, including embryonic development, bone formation, and cancer. The manuscript highlights the potential therapeutic implications of BMP signaling in cancer treatment, particularly in glioblastoma, breast cancer, and other malignancies. The discussion on BMP inhibitors and their possible use in therapy is timely and relevant. The manuscript cites recent studies (up to 2024), ensuring that the review is current and reflects the latest advancements in the field.
Response 2: We thank the reviewer for your thoughtful and encouraging feedback.
Comments 3: Weaknesses: The manuscript primarily summarizes existing studies without providing sufficient critical analysis or synthesis of the findings. For example, the contradictory roles of BMPR2 in breast cancer (lines 198-201) could be discussed in greater depth, with potential explanations for these discrepancies (e.g., differences in experimental models, cell types, or signaling contexts).
Response 3: We thank the reviewer for their valuable suggestions. We agree with this comment. As pointed out we have incorporated the potential explanations for the discrepancies in role of BMPR2 in breast cancer and highlighted in revised manuscript. Page No:7 and 8, Paragraph: 18 and 19, Line No: 266 – 292.
Comments 4: The manuscript could benefit from discussing the challenges and limitations of targeting BMP signaling in cancer therapy. For example, the potential for off-target effects or resistance mechanisms could be explored.
Response 4: We thank the reviewer for their comment. We appreciate your suggestion and agree that discussing the challenges and limitations of targeting BMP signaling in cancer therapy would enhance the manuscript. We have incorporated a section addressing potential off-target effects, resistance mechanisms, and other limitations to provide a more balanced perspective on BMP-based therapeutic strategies and highlighted in revised manuscript. Page No: 11, Paragraph: 35, 36 and 37, Line No: 438 – 464.
Comments 5: Introduction The introduction could be strengthened by briefly mentioning the clinical significance of BMP signaling, setting the stage for the later discussion on cancer.
Response 5: Thank you for your suggestion. We have revised this section to briefly introduce the role of BMP signalling in disease progression and potential therapeutic applications, ensuring a stronger connection to the subsequent discussions and highlighted in revised manuscript. Page No: 2, Paragraph: 5, Line No: 77 – 94.
Comments 6: Line 110 to line 118, BMPR2 and BMP2R are mixed used.
Response 6: Thank you for the suggestion. We have made the correction in the revised manuscript and highlighted in revised manuscript. Page No: 4, Paragraph: 10, Line No 157 and 161.
Comment 7: BMP Receptor Signaling (Section 1.2): The discussion of non-SMAD signaling (lines 90-93) could be expanded to include more details on the downstream effects of these pathways. ID1 is downstream of BMP signaling; however, other than ID1, other targets could be discussed in some tissue, for example, bone tissue: Msx2.
Response 7: We thank the reviewer for this important suggestion. We appreciate your suggestion to expand the discussion on non-SMAD signaling and its downstream effects. We have expanded the discussion on non-SMAD signaling and also have added other targets like Tlx-2, Msx2 and Dlx5. Muscle segment homeobox (Msx2), transcription factor with a homeobox, is a member of the highly conserved and extensively expressed msh superfamily. BMP signaling actively modulates the expression of Msx2 in numerous developmental processes like osteogenesis, odontogenesis, and limb patterning. Dlx5 is a transcription factor that promotes bone formation and is expressed during the later stages of osteoblast differentiation. Tlx-2, a homobox gene, expressed in mouse embryos' primitive streak necessary for gastrulation. We have incorporated the changes and highlighted in revised manuscript. Page No: 3, Paragraph: 8, Line No: 118 – 139.
Comment 8: BMPR2 in Cancer (Section 1.4.3): The contradictory findings regarding BMPR2 in breast cancer (lines 198-201) should be analyzed in greater depth, with potential explanations for these discrepancies.
Response 8: We thank the reviewer for their valuable suggestions. As pointed out we have incorporated the potential explanations for the discrepancies in role of BMPR2 in breast cancer and highlighted in revised manuscript. Page No: 7 and 8, Paragraph: 18, Line No: 266 – 292.

Reviewer 2 Report
Comments and Suggestions for Authors
Dear authors,
I evaluated the article titled “A comprehensive review exploring the role of bone morphogenetic proteins [BMP]: Biological mechanisms”.
Although the topic is interesting, this study did not find any novelty or deep knowledge, and I considered it incomplete. The authors discussed its role in osteogenesis and cancer, but it could have been better focused.
Moreover, one important article was published showing BMP (rh) and osseointegration.
I suggest to include it.
(new ref) The Influence of rhBMP-7 Associated with Nanometric Hydroxyapatite Coatings Titanium Implant on the Osseointegration: A Pre-Clinical Study. Polymers. 2022, 14(19), 4030; https://doi.org/10.3390/polym14194030
It did not show the technological development of recombinant human (rh) BMP. It is important for this type of study.
The abstract must be improved. Include the objective; present a better development of results.
Present the goal of the study.
The general presentation does not follow the basic points of a scientific article. For example, there is no methodology to clarify how the steps were developed.
The authors gathered a good level of information and are presenting them. But it should have more focus.
Author Response
Comments 1: Dear authors, I evaluated the article titled “A comprehensive review exploring the role of bone morphogenetic proteins [BMP]: Biological mechanisms”. Although the topic is interesting, this study did not find any novelty or deep knowledge, and I considered it incomplete. The authors discussed its role in osteogenesis and cancer, but it could have been better focused.
Response 1: Thank you for taking the time to evaluate our manuscript and for your constructive feedback. We appreciate your insights and understand your concerns regarding the depth and focus of our review. BMPs function as both stimulators and inhibitors in many pathways and process of development; hence, it is inappropriate to categorically classify BMPs as either oncogenes or anti-oncogenes. The aforementioned findings collectively revealed that the impact of BMP signaling on tumor progression is contingent upon the cell types and the tumor microenvironment. Therefore, our goal was to provide a broad yet informative overview of BMPs and their biological mechanisms, particularly BMPR2 in vascular development, osteogenesis and cancer.
Comments 2: Moreover, one important article was published showing BMP (rh) and osseointegration. I suggest to include it. (new ref) The Influence of rhBMP-7 Associated with Nanometric Hydroxyapatite Coatings Titanium Implant on the Osseointegration: A Pre-Clinical Study. Polymers. 2022, 14(19), 4030; https://doi.org/10.3390/polym14194030. It did not show the technological development of recombinant human (rh) BMP. It is important for this type of study.
Response 2: Thank you for the suggestion. We have included the article as per suggested and highlighted in revised manuscript. Page No: 5, Paragraph: 12, Line No: 187 – 203.
Comments 3: The abstract must be improved. Include the objective; present a better development of results.
Response 3: We express our gratitude to the reviewer for their insightful recommendations. The abstract has been improved by stating the objective and providing results in a more concise manner and highlighted in revised manuscript. Page No: 1, Line No: 12 - 24
Comments 4: Present the goal of the study.
Response 4: We thank the reviewer for their comment. The goal of the study is to offer a thorough examination of the function of Bone Morphogenetic Proteins (BMPs) and BMP receptor signaling, with a specific emphasis on BMPR2 and highlighted in revised manuscript. Page No: 2, Paragraph: 3,5,8, Line No: 60 to 62, 91 to 94, 137 to 139.
Comments 5: The general presentation does not follow the basic points of a scientific article. For example, there is no methodology to clarify how the steps were developed. The authors gathered a good level of information and are presenting them. But it should have more focus.
Response 5: We thank the reviewer for the suggestion. But the article is a comprehensive review that aims to provide a broad and in-depth discussion of a BMPs by synthesizing existing literature. This comprehensive review does not follow a rigid, pre-defined search strategy or data extraction process. Instead, it provides a thematic discussion based on expert knowledge and existing literature. Additionally, this comprehensive article is structured in accordance with the journal's criteria.

Round 2
Reviewer 1 Report
Comments and Suggestions for Authors
The manuscript provides a comprehensive review of the role of Bone Morphogenetic Proteins (BMPs) and BMP receptor signaling, with a particular focus on BMPR2. The review is well-structured and covers a broad range of topics, making this manuscript a valuable resource for biology researchers.
The modified version addressed my comments well. It can be accepted.
Reviewer 2 Report
Comments and Suggestions for Authors
Dear Authors,
I appreciate the responses and adjustments.
Congratulations.